# Albumin change predicts failure in ulcerative colitis treated with adalimumab

**Natsuki Ishida[1], Kenichi Takahashi[2], Yusuke Asai[2], Takahiro Miyazu[2], Satoshi Tamura[2], Shinya Tani[2], Mihoko Yamade[2], Moriya Iwaizumi[3], Yasushi Hamaya[2], Satoshi Osawa[1], Ken Sugimoto[2]\***

1 Department of Endoscopic and Photodynamic Medicine, Hamamatsu University School of Medicine, Hamamatsu, Shizuoka, Japan, 2 First Department of Medicine, Hamamatsu University School of Medicine, Hamamatsu, Shizuoka, Japan, 3 Department of Laboratory Medicine, Hamamatsu University School of Medicine, Hamamatsu, Shizuoka, Japan

\* sugimken@hama-med.ac.jp

**Data Availability Statement:** All data needed to evaluate the conclusions in the paper are presented herein. Additional data related to this study may be requested from the authors.

## Abstract

Anti-tumor necrosis factor (TNF) -α antibodies, including infliximab (IFX), adalimumab (ADA), and golimumab, which were the first biologic therapeutic agents, have a crucial position in advanced therapy for ulcerative colitis (UC). We aimed to investigate serum albumin (Alb) change as a prognostic factor for the therapeutic effect of ADA in UC. Thirty-four patients with UC treated with ADA were enrolled in this study and were divided into failure and non-failure groups. Biological data, such as Alb were compared between the two groups. Thirteen patients showed failure within six months. Examination of the biological data showed a significant difference between the two groups only in the week 2/week 0 Alb ratio. In receiver-operating characteristic (ROC) curve analysis to predict failure, the cut-off value of week 2/week 0 Alb ratio was 1.00, and the area under the curve was 0.868 (95% confidence interval: 0.738–0.999). In addition, in the sub-group analysis of only clinically active patients, the week 2/week 0 Alb ratio of the non-failure group was significantly higher than that of the failure group, and the cut-off-value in ROC analysis was 1.00. Week 2/week 0 Alb ratio ≤ 1 predicts failure within six months of ADA for UC.

## Introduction

Various biological agents have emerged for the treatment of ulcerative colitis (UC). Anti-tumor necrosis factor (TNF)-α antibodies, including infliximab (IFX), adalimumab (ADA), and golimumab, which were the first biologic therapeutic agents, have a crucial role in the advanced therapy for UC [1–4]. Following IFX, ADA is an anti-TNF-α antibody agent administered by subcutaneous injection, which is indicated for the treatment of UC [2, 3, 5]. The ULTRA-1/2 trial reported its usefulness in the induction of remission, and although there were differences in the extent of previous use of molecular-targeted drugs, the clinical remission rate at week 8 was 16.5%–18.5%, showing good results [2, 3]. The subsequent ULTRA-3 trial was an unblinded clinical study in which only patients who completed the ULTRA-1/2 studies were included, and a clinical remission rate of 24.7% was reported after four years [5]. Thus, large-scale clinical trials have reported the efficacy of ADA from induction to maintenance therapy.

**Funding:** The funders had no role in study design, data collection and analysis, decision to publish, or preparation of the manuscript.

**Competing interests:** The authors have declared that no competing interests exist.

Although the efficacy of anti-TNF-α antibody preparations has been reported, there are cases of primary failure, in which treatment response is not obtained, and secondary failure, in which treatment resistance occurs even after temporary improvement [6, 7]. Although there are various reports on the evaluation methods, the frequency of primary failure of anti-TNF-α antibody preparations in inflammatory bowel disease (IBD) is approximately 10%–40% overall [6]. In addition, reports on UC indicate that the rate of primary failure is approximately 30%–40% [6]. The frequency of secondary failure of ADA in UC has been reported to be 58.3% [7].

Predictive factors for primary and secondary failure have been reported [8–11]. A retrospective cohort study based on data from the ENEIDA registry reported that failure to achieve response at 12 weeks was a clinical predictor of colectomy [8]. A randomized controlled trial reported that high C-reactive protein (CRP) levels, primary sclerosing cholangitis comorbidity, and female sex were predictors of failure [9]. Previous biologic exposure and disease activity at eight weeks after induction were also reported to be predictors of ADA [10]. The mechanism of developing such a loss of response (LOR) is involved in the production of anti-drug antibodies and the decrease in the serum level (trough level) of anti-TNF-α due to the undetectability of anti-TNF-α; both serum ADA trough level and anti-ADA antibodies (AAA) have been reported to be associated with mucosal healing and LOR [11]. Various prognostic factors for ADA in UC treatment have been reported. Previously, Lee et al. reported that the ratio of albumin (Alb) during induction and after two weeks of anti-TNF-α antibody therapy in UC was a predictor of subsequent prognosis [12]. In this study, anti-TNF-α agents, including both IFX and ADA, were evaluated, but no study has examined the Alb ratio only for ADA. In this study, we evaluated the effect of ADA on UC and examined whether the Alb ratio is useful for prognosis prediction.

## Methods

### Patients and study design

Patients with UC who were treated with ADA at the Hamamatsu University School of Medicine between November 2013 and February 2022 were enrolled in this study. This study protocol was reviewed and approved by the Ethics Committee of Hamamatsu University School of Medicine (number 21–029). This study was conducted in accordance with Good Clinical Practice principles in adherence to the Declaration of Helsinki. Informed consent for patients was obtained in the form of opt-out on the hospital website. The collection and analysis of data began in October 17, 2022. We did not have access to information that could identify individual participants during or after data collection.

The patients were diagnosed with UC according to the current established UC criteria, excluding IBD such as indeterminate colitis or inflammatory bowel disease-unclassified [13]. The primary endpoint of this retrospective, single-center, observational study was the continuation of treatment with ADA, which did not require intensification of treatment. We defined failure as any of the following with UC progression: change from ADA to another agent; surgery, induction, or ADA doubling was performed; or treatment with prednisolone (PSL) or tacrolimus (TAC). The secondary endpoint compared failure and non-failure only in clinically active patients. Non-failure patients were followed up for at least six months, and patients who dropped out for reasons other than failure were excluded. Based on these investigations, we searched for factors that could predict the therapeutic effect of ADA treatment.

### Disease assessment

In this study, the clinical activity index (CAI) according to Rachmilewitz was used to evaluate the clinical activity of UC [14]. Clinical remission was defined as CAI 4, while the clinical

response was defined as a decrease of more than 1 point compared to the baseline and a decrease of 50% from the baseline. Biological data including serum albumin (Alb), serum CRP, white blood cell (WBC), hemoglobin (Hb), and platelet (Plt) levels, were measured at the laboratory test department of Hamamatsu University School of Medicine. For all enrolled patients, these data were measured and evaluated at induction and two weeks later. Endoscopic examination of UC severity was assessed by the Mayo endoscopic subscore (MES) and the UC endoscopic index of severity (UCEIS) [15, 16].

### Treatment and follow-up of patients

Patients enrolled in this study visited our hospital regularly every one week to two months. ADA was administered by subcutaneous injection of 160 mg during induction, 80 mg after two weeks, and 40 mg every two weeks from the fourth week onwards. We instructed patients to record their clinical symptoms based on the CAI (Rachmilewitz index) to assess their clinical activity outside the hospital. Based on the exacerbation of clinical symptoms, such as an increase in the number of defecations and the appearance of bloody stools, the treatment policy was decided based on the judgment of the attending physician.

### Statistical analysis

Statistical analyses were performed with IBM SPSS Statistics for Windows, version 24 (IBM Corp., Armonk, N.Y., USA) and EZR (Saitama Medical Center, Jichi Medical University, Saitama, Japan) software [17]. The Mann–Whitney U test or Student's t-test was used to evaluate differences. The cumulative non-failure rate was evaluated using Kaplan–Meier analysis with the log-rank test. $P < 0.05$ was considered statistically significant.

## Results

### Patient characteristics

Forty patients with UC treated with ADA were enrolled at our institution (Fig 1). Of these patients, four were lost to follow-up within six months, and two discontinued treatment owing to adverse events; thus, a total of six patients were excluded. The remaining 34 were included in the analysis (Table 1). The median age of the patients in the study was 43 years, and the median disease duration was seven years. There were 27 patients (79.4%) with extensive colitis, 6 (17.6%) with left-sided colitis, and 1 (2.9%) with proctitis. In the first investigation of this study, all patients were enrolled regardless of UC activity, including six patients with MES 0 or 1 and mucosal healing. These patients included those who switched from other treatments due to adverse effects and those who were induced as maintenance therapy after remission was induced with PSL or TAC. Seven (20.6%) patients had a history of using biologics.

### Comparison of failure and non-failure with ADA treatment

Thirteen patients failed during the six-month follow-up period (Table 2). The patient background during entry was compared between the failure and non-failure groups, but no items showed a significant difference. Then, we compared the biological data between the failure and non-failure groups (Table 3). Biological data in weeks 0 and 2 did not show a significant difference between the two groups. When the ratio of weeks 0 and 2 was calculated for each biological data and compared, the week 2/week 0 Alb ratio was significantly higher in the non-failure group than in the failure group ($P < 0.001$).

Furthermore, the actual number, proportion, and ratio of leukocyte fractions were compared between the two groups, but no variables showed significant differences (S1 Table).

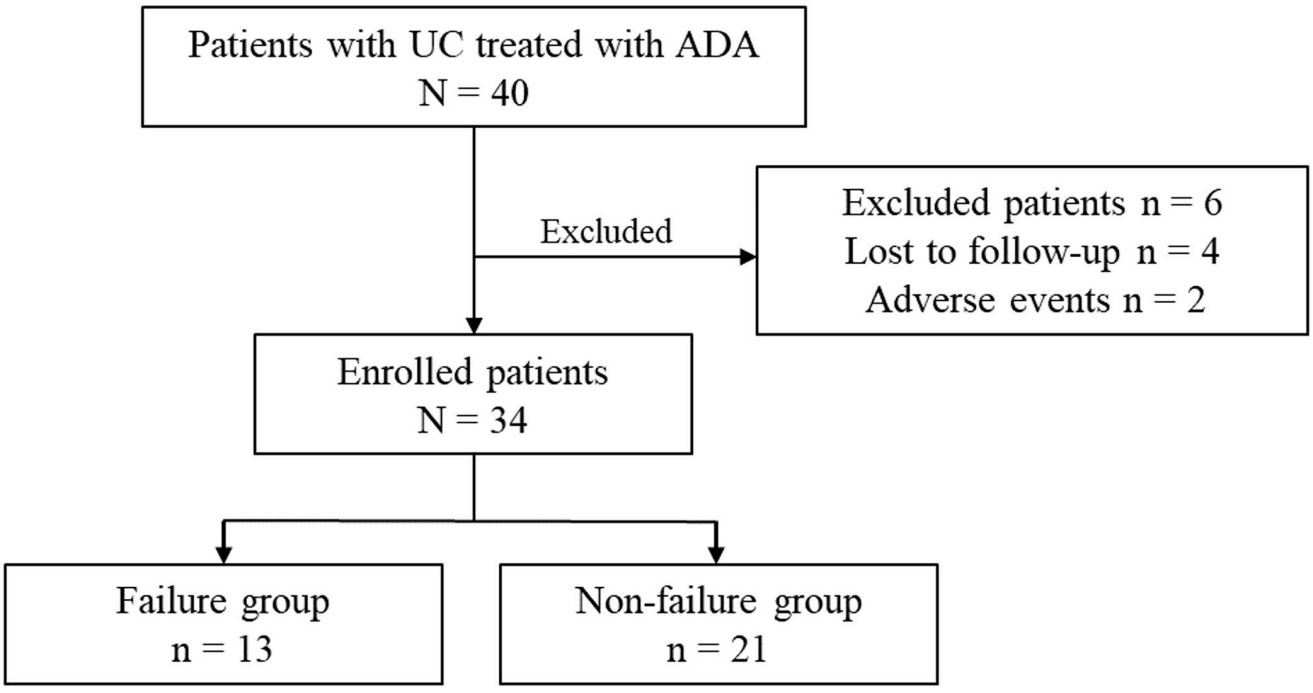

**Fig 1. Study flow chart.** There were 40 patients with ulcerative colitis (UC) treated with adalimumab (ADA). Six patients were excluded, and the remaining 34 patients were enrolled in this study. There were 13 and 21 patients in failure and non-failure groups, respectively.

Although the sample size decreased from that at baseline, no biologicals data showed a significant difference between the failure and non-failure groups at week 6 (S2 Table).

## Prediction of failure within six months using Alb ratio in ADA treatment

We performed receiver-operating characteristic (ROC) curve analysis to predict failure within six months using the week 2/week 0 Alb ratio (Fig 2a). The cut-off value of the Alb ratio for predicting failure was 1. The area under the curve (AUC) for this analysis was 0.868 (95% confidence interval 0.738–0.999). Next, Kaplan-Meier analysis was performed by dividing into week 2/week 0 Alb ratio ≥ 1.00 group and week 2/week 0 Alb ratio < 1.00 group; a significant difference was shown in the log-rank test (P < 0.001) (Fig 2b). In addition, multivariate analysis showed that week 2/week 0 Alb ratio < 1.00 was an independent prognostic factor for predicting failure within six months (Table 4).

## Examination of the week 2/week 0 Alb ratio in clinically active patients

Analysis up to the above was performed on all enrolled patients regardless of CAI, and then we analyzed the week 2/week 0 Alb ratio of 25 patients with clinical activity excluding nine patients in clinical remission with CAI ≤ 4. There was no significant difference in the baseline patient background data between the group of 10 failures and the group of 25 non-failures (S3 Table). The week 2/week 0 Alb ratio showed a significant difference between the failure and non-failure groups (P = 0.002) (Fig 3a). The ROC curve analysis for predicting failure within six months had a cut-off value of 1 and an AUC of 0.86 (95% CI 0.698–1.000), similar to the analysis of all patients (Fig 3b). In addition, when Kaplan–Meier analysis was performed on

**Table 1. Baseline patient characteristics.**

| Characteristics at entry | | N = 34 |
|---|---|---|
| Age (year), median [IQR] | | 43 [25–55] |
| Male/Female, n (%) | | 18 (52.9) /16 (47.1) |
| Disease duration (year), median [IQR] | | 7 [3–15] |
| Disease extent, n (%) | Extensive colitis | 27 (79.4) |
| | Left-sided colitis | 6 (17.6) |
| | Proctitis | 1 (2.9) |
| CAI (Rachmilewitz index), median [IQR] | | 7 [4–10] |
| Alb (g/dL), median [IQR] | | 3.7 [3.3–4.1] |
| CRP (mg/dL), median [IQR] | | 0.30 [0.10–1.64] |
| WBC (/µL), median [IQR] | | 7755 [5385–9907] |
| Hb (g/dL), median [IQR] | | 12.4 [10.0–14.2] |
| Plt (×10$^4$/µL), median [IQR] | | 37.9 [28.6–41.7] |
| MES, n (%) | MES 0 | 1 (2.9) |
| | MES 1 | 5 (14.7) |
| | MES 2 | 17 (50.0) |
| | MES 3 | 11 (32.4) |
| UCEIS, median [IQR] | | 4 [3–5] |
| Other medication, n (%) | Oral 5-ASA | 18 (52.9) |
| | Suppository steroids | 2 (5.9) |
| | Systemic steroids | 14 (41.2) |
| | Immunomodulator | 11 (32.4) |
| | Tacrolimus | 3 (8.8) |
| History of biologics use | | 7 (20.6) |

IQR, interquartile range; CAI, clinical activity index; Alb, albumin; CRP, C-reactive protein; WBC, white blood cell; Hb, hemoglobin; Plt, platelet; MES, Mayo endoscopic subscore; UCEIS, ulcerative colitis endoscopic index of severity; 5-ASA, 5-aminosalicylic acid

groups with a week 2/week 0 ≥ 1 and < 1 in this sub-group analysis, a significant difference was shown in the log-rank test (P = 0.002) (Fig 3c).

## Examination of the week 2/week 0 Alb ratio in patients treated with anti-tumor necrosis factor-α antibodies

In addition to the 34 patients in the ADA group, 20 and 14 patients treated with GLM and IFX, respectively, were added, for a total of 68 patients. As a statistically significant analysis cannot be performed due to the small sample size, analysis was not performed for the GLM and IFX group. During the 6-month observation period, 20 patients had treatment failure. There were no baseline characteristics that showed significant differences between the failure and non-failure groups (S4 Table). Meanwhile, the week 2/week 0 Alb ratio showed a significant difference between the failure group and the non-failure groups (P < 0.001), and the ROC analysis for predicting treatment failure had a cut-off value of 0.98 and an AUC of 0.84 (95% CI: 0.727–0.960) (Fig 4a, 4b). Kaplan–Meier analysis was also performed for the week 2/week 0 Alb ratio ≥0.98 and <0.98 groups, and the log-rank test showed a significant difference (P < 0.001) (Fig 4c).

**Table 2. Comparison of patient characteristics between the failure and the non-failure groups.**

| Characteristics at entry | | Failure | Non-failure | P-value |
|---|---|---|---|---|
| | | n = 13 | n = 21 | |
| Age (year), median [IQR] | | 36 [23–62] | 47 [32–54] | 0.375 |
| Male/Female, n (%) | | 6 (46.2) /7 (53.8) | 12 (57.1) /9 (42.9) | 0.725 |
| Disease duration (year), median [IQR] | | 5 [1–10] | 9 [4–15] | 0.136 |
| Disease extent, n (%) | Extensive colitis | 10 (76.9) | 17 (81.0) | 0.789 |
| | Left-sided colitis | 3 (23.1) | 3 (14.3) | |
| | Proctitis | 0 (0.0) | 1 (4.8) | |
| CAI (Rachmilewitz index), median [IQR] | | 8 [5–10] | 7 [4–10] | 0.749 |
| MES, n (%) | MES 0 | 1 (7.7) | 0 (0.0) | 0.543 |
| | MES 1 | 2 (15.4) | 3 (14.3) | |
| | MES 2 | 5 (38.5) | 12 (57.1) | |
| | MES 3 | 5 (38.5) | 6 (28.6) | |
| UCEIS, median [IQR] | | 4 [2–6] | 5 [3–5] | 0.957 |
| Other medication, n (%) | Oral 5-ASA | 6 (46.2) | 12 (57.1) | 0.725 |
| | Suppository steroids | 0 (0.0) | 2 (9.5) | 0.513 |
| | Systemic steroids | 5 (38.5) | 9 (42.9) | 1 |
| | Immunomodulator | 4 (30.8) | 7 (33.3) | 1 |
| | | 1 (7.7) | 2 (9.5) | 1 |
| History of biologics use | | 3 (23.1) | 4 (19.0) | 1 |

IQR, interquartile range; CAI, clinical activity index; Alb, albumin; CRP, C-reactive protein; WBC, white blood cell; Hb, hemoglobin; Plt, platelet; MES, Mayo endoscopic subscore; UCEIS, ulcerative colitis endoscopic index of severity; 5-ASA, 5-aminosalicylic acid

**Table 3. Comparison of biological data between the failure and the non-failure groups.**

| Variable | Failure | Non-failure | P-value |
|---|---|---|---|
| | n = 13 | n = 21 | |
| Alb at week 0 (g/dL), median [IQR] | 3.8 [3.7–3.9] | 3.6 [3.3–4.3] | 0.749 |
| CRP at week 0, median [IQR] | 0.22 [0.06–2.24] | 0.32 [0.13–1.05] | 0.958 |
| WBC at week 0 (/μL), median [IQR] | 7760 [5110–9530] | 7750 [5610–10080] | 0.901 |
| Hb at week 0 (g/dL), median [IQR] | 12.80 [10.0–14.4] | 11.8 [10.3–13.6] | 0.670 |
| Plt at week 0 (×104/μL), median [IQR] | 38.1 [30.9–42.3] | 37.8 [28.3–41.6] | 0.818 |
| Alb at week 2 (g/dL), median [IQR] | 3.5 [3.2–3.9] | 4.0 [3.6–4.5] | 0.058 |
| CRP at week 2, median [IQR] | 0.22 [0.03–0.53] | 0.08 [0.02–0.22] | 0.243 |
| WBC at week 2 (/μL), median [IQR] | 6240 [4760–9850] | 7410 [4360–8450] | 0.790 |
| Hb at week 2 (g/dL), median [IQR] | 12.3 [10.4–13.1] | 12.4 [10.4–13.4] | 0.696 |
| Plt at week 2 (×104/μL), median [IQR] | 36.3 [24.4–46.9] | 32.5 [24.8–42.9] | 0.490 |
| Week 2/week 0 Alb ratio | 0.95 [0.91–1] | 1.06 [1.03–1.16] | < 0.001 |
| Week 2/week 0 CRP ratio | 0.55 [0.30–1] | 0.44 [0.10–0.80] | 0.446 |
| Week 2/week 0 WBC ratio | 0.89 [0.75–1.02] | 0.80 [0.73–0.98] | 0.736 |
| Week 2/week 0 Hb ratio | 1.01 [0.97–1.04] | 1.05 [0.98–1.07] | 0.184 |
| Week 2/week 0 Plt ratio | 1.04 [0.78–1.13] | 0.98 [0.79–1.03] | 0.559 |

Alb, albumin; IQR, interquartile range; CRP, C-reactive protein; WBC, white blood cell; Hb, hemoglobin; Plt, platelet

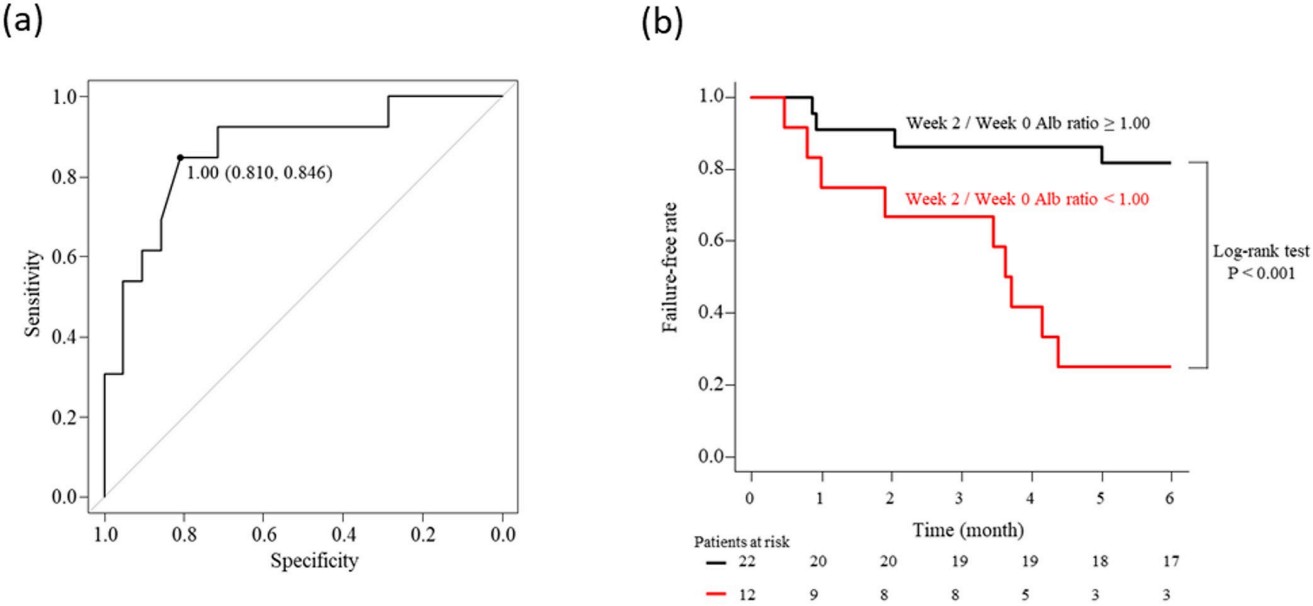

**Fig 2.** Receiver-operating characteristic (ROC) curve analysis for prediction of failure by treatment with adalimumab to patients with ulcerative colitis during six months (a) and Kaplan–Meier analysis of failure-free rate by the groups of patients with week 2/week 0 Alb ratio $\geq$ 1.00 and < 1.00 (b).

## Discussion

The turning point in the treatment of UC was the emergence of biological agents, and anti-TNF-α antibody agents had a particularly large impact. ADA, an anti-TNF-α antibody preparation, was shown to be useful in ULTRA-1/2/3 studies [2, 3, 5]. However, ADA is a drug that induces anti-drug antibodies upon administration, and the emergence of drug antibodies is considered to lead to a loss of response to ADA. Therefore, there are reports that anti-ADA antibodies and measurement of serum ADA levels may be effective predictors of ADA efficacy [11, 18]. However, despite studies involving Crohn's disease (CD) and UC, we reported that anti-ADA antibodies did not correlate with treatment adherence, and the prognostic utility of anti-ADA antibodies is controversial [19]. In addition, the measurement of these anti-ADA antibodies and ADA trough concentrations is problematic as they cannot be measured easily due to the difficulty of the measurement method and the cost of measurement. Moreover,

**Table 4. Multivariate analysis for predicting failure within 6 months.**

| | Univariate analysis | | | Multivariate analysis | | |
|---|---|---|---|---|---|---|
| | HR | 95% CI | P-value | HR | 95% CI | P-value |
| Week 2/week 0 Alb ratio < 1.00 | 6.248 | 1.889–20.66 | 0.003 | 16.62 | 3.135–88.08 | 0.001 |
| Age | 0.984 | 0.951–1.019 | 0.366 | 0.966 | 0.918–1.017 | 0.19 |
| Sex, male | 0.674 | 0.226–2.006 | 0.478 | 0.629 | 0.154–2.564 | 0.518 |
| Disease duration | 0.957 | 0.884–1.035 | 0.271 | 0.986 | 0.910–1.068 | 0.723 |
| Extensive colitis | 0.827 | 0.227–3.008 | 0.773 | 0.187 | 0.032–1.102 | 0.064 |
| CAI $\geq$ 5 | 1.256 | 0.345–4.569 | 0.729 | 3.191 | 0.369–27.58 | 0.292 |
| MES $\geq$ 2 | 0.670 | 0.184–2.443 | 0.544 | 0.730 | 0.096–5.568 | 0.762 |
| History of biologics use | 1.144 | 0.315–4.162 | 0.838 | 4.200 | 0.648–27.24 | 0.132 |

HR, hazard ratio; 95% CI, 95% confidence interval; Alb, albumin; CAI, clinical activity index; MES, Mayo endoscopic subscore

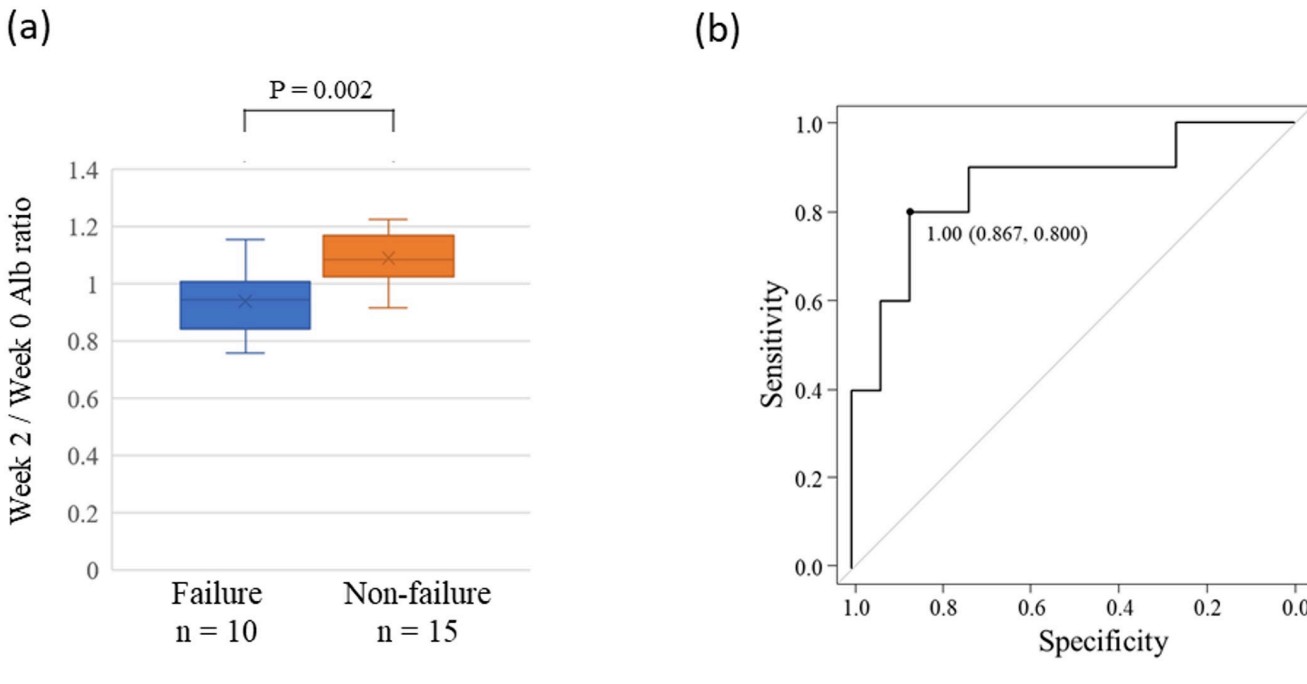

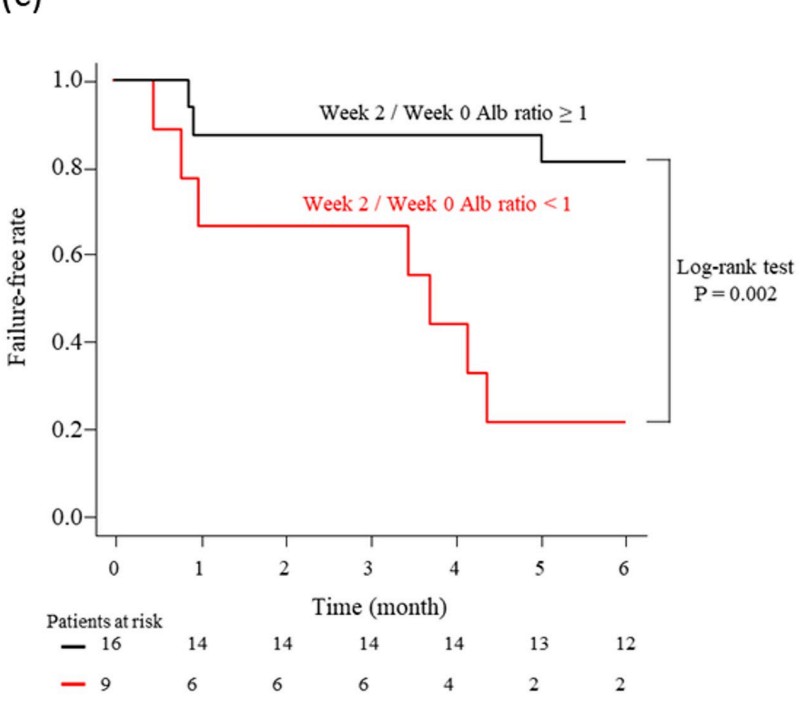

**Fig 3.** Week 2/week 0 Alb ratio difference between the patients with failure and non-failure in the group with CAI > 4 (a). Receiver-operating characteristic curve analysis for prediction of failure by treatment with adalimumab to patients with ulcerative colitis during six months in the group with CAI > 5 (b). Kaplan–Meier analysis of failure-free rate by the groups of patients with week 2/week 0 Alb ratio ≥ 1.00 and < 1.00 (c).

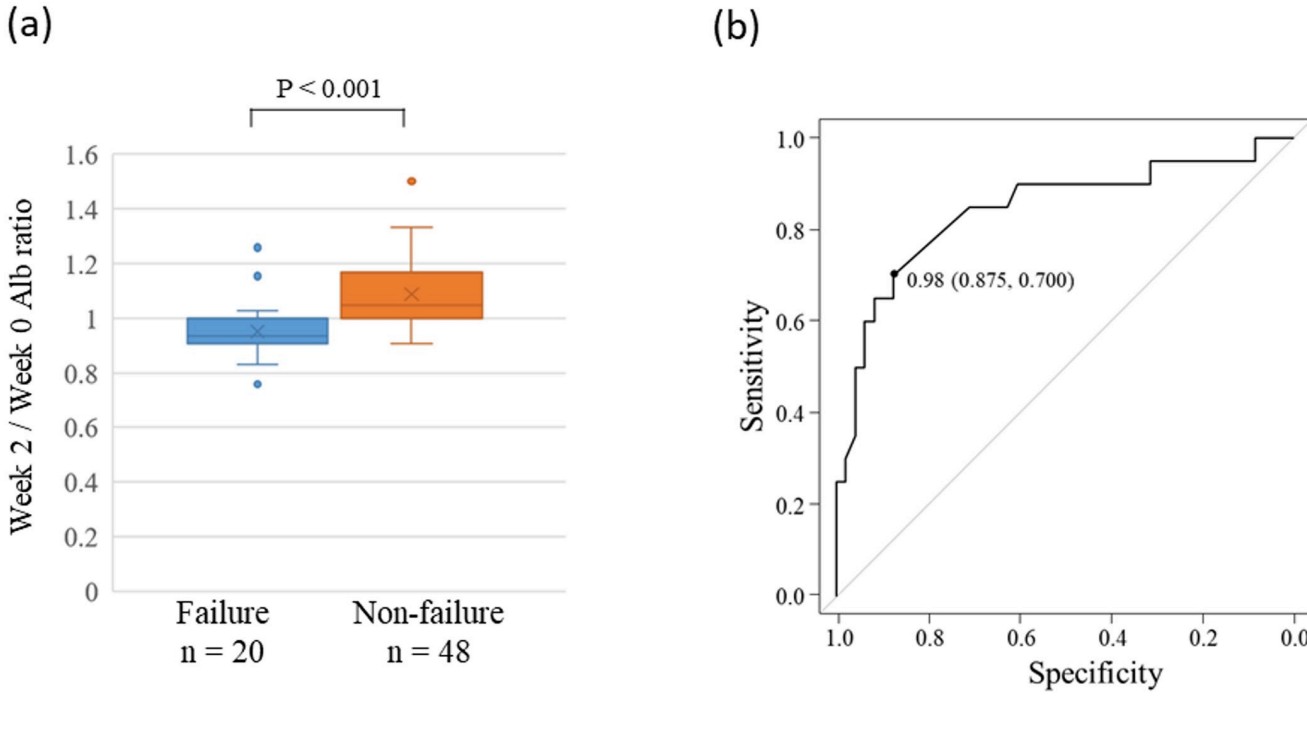

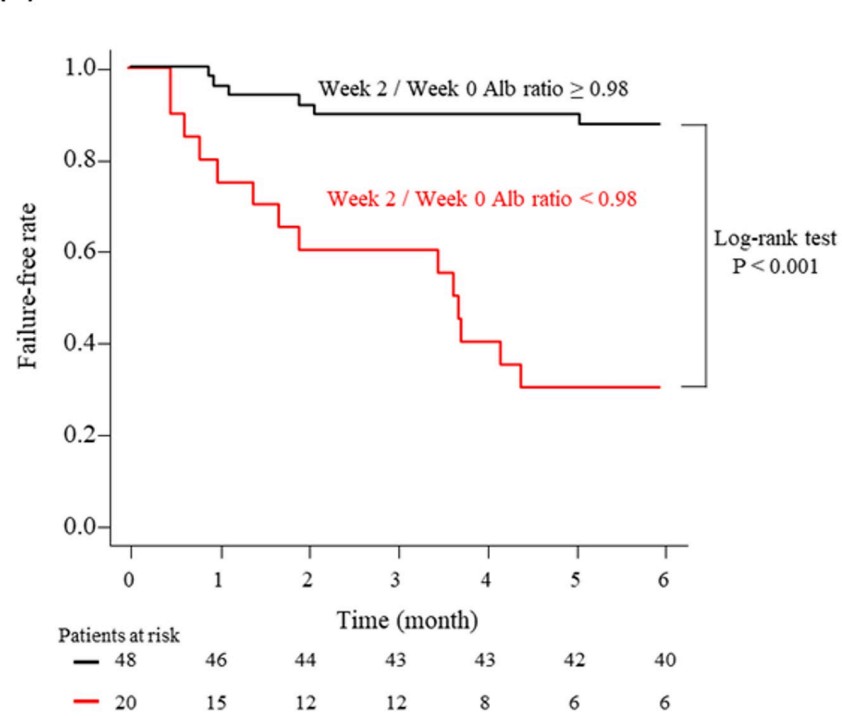

**Fig 4.** Difference in the week 2/week 0 Alb ratio of patients treated with anti-tumor necrosis factor (TNF)-α antibodies in the failure and non-failure groups (a). Receiver-operating characteristic curve analysis for predicting treatment failure within 6 months following administration of anti-TNF-α to patients with ulcerative colitis (b). Kaplan–Meier analysis of the failure-free rate among patients with a week 2/week 0 Alb ratio of ≥0.98 and <0.98 (c).

ADA was a drug that appeared in the first era of biologics therapy, and there were almost no therapeutic drugs as advanced therapy to compensate in case of failure. Therefore, evaluation of efficacy is crucial, and there have been reports of evaluating efficacy by measuring anti-ADA antibody and ADA trough concentration. However, in recent years, a large number of drugs, mainly biologics, have been developed for UC, and even when a failure occurs, the transition to the next treatment is easier than before. Further, the blood concentration and anti-drug antibody measurements for predicting the efficacy of ADA have declined. However, ADA is still a crucial biologic drug in terms of its efficacy and is the only anti-TNFα antibody drug that can be increased and shortened in the dose for UC.

Prediction of treatment efficacy in UC is important. A universal efficacy predictor that can be assessed at any facility is desired. Lee et al. reported that anti-TNF-α antibody preparations, including IFX and ADA, showed that the Alb ratio during the induction and at two weeks predicted relapse [12]. However, both drugs were summarized in the report, and ADA alone was not evaluated. In this study, we focused only on ADA and examined whether it can be used to predict the prognosis. The Alb ratio was found to be useful as a predictor even with ADA alone. Conversely, it would have been desirable to examine the usefulness of the Alb ratio in predicting the efficacy of other anti-TNF-α antibodies, including IFX and GLM; however, the sample size was small, and analysis was not possible. It is hoped that IFX and GLM will be investigated based on accumulated results in future cases. It was also reported that the neutrophil to lymphocyte ratio is useful for predicting the efficacy of anti-TNF-α antibody drugs [20]. In this study, leukocyte differential analysis was performed, but no significant differences were shown for any of the variables, which may be due to the small sample size and differences in endpoints. In the first analysis of this study, all ADA-treated patients with UC were evaluated regardless of clinical activity as real-world data. In other words, the study included remission cases such as patients who were introduced to maintenance therapy after induction of remission with PSL or TAC and patients who were switched due to adverse events of other treatments. Although the inclusion of these patients has the advantage of being real-world data in clinical practice, it is problematic that the inclusion of patients in remission is fraught with bias when evaluating the remission induction effects of ADA. Therefore, we performed a second analysis of clinically active patients, excluding those with CAI (Rachmilewitz index) $\leq 4$. As a result, the Alb ratio showed a significant difference between the failure and non-failure groups. The cut-off value in the ROC analysis was 1, which was the same result as the analysis of all patients.

The cut-off value of the Alb ratio obtained in this study was 1, and this cut-off value is easy to interpret. If the Alb value at week 2 is higher than that at the time of induction, the subsequent prognosis is favorable. This result is a subsequent predictive factor and representative of the early treatment effects of ADA at week 2. In a previous study, we reported that the week 2/week 0 Alb ratio could be a prognostic factor for TAC as well, and the cut-off value was 1 [21]. However, a subsequent study reported that the week 2/week 1 Alb ratio could be a more accurate prognostic predictor [22]. This is because TAC, which is mostly induced in inpatient and biological data, can be easily obtained one week after its induction, whereas ADA is often induced in outpatient treatment and is usually measured two weeks after its induction. Therefore, it was difficult to obtain week 1 data. Although there are some reports on the prognosis and Alb of IBD treatment using anti-TNFα antibody preparations, Tighe et al. showed no significant difference between failure and non-failure Alb levels at the time of induction [23]. Conversely, Baki et al. reported that the Alb level during induction of the anti-TNF-α antibody preparation in the group that did not achieve remission was significantly lower than that in the group that achieved remission [24]. There are differences in the endpoint, and there are various reports on Alb at the time of induction and prognosis; however, this study also did not

show a significant difference in the Alb value during induction. Hence, we concluded that the improvement rate of Alb at 2 weeks affected the prognosis. As an additional analysis, we performed an analysis using all currently available anti-TNF-α antibody preparations, including patients treated with GLM and IFX, in addition to ADA. The results were useful for prediction. Furthermore, it would be sufficient to perform similar analyses with GLM or IFX alone, and it is hoped that future cases will accumulate.

There are some limitations to this study. First, this was a single-institution study. The number of registered patients was small. Second, this was retrospective research. Third, there was no investigation of biomarkers and histological evaluation before and after the treatment. Although treatment in recent years aiming at treat-to-target has become the standard for UC, the achievement of mucosal healing was not evaluated, and the focus was only on the clinical activity in this study. In addition, the endpoint of this study was a point that was interpreted for our convenience. However, although the number of participants in this study is small, it is a strength that the week 2/week 0 Alb ratio showed a significant difference between the failure and non-failure groups, which is a useful result for predicting clinical prognosis.

## Conclusion

Week 2/week 0 Alb ratio was a useful prognostic predictor in ADA treatment in UC. If the Alb value at week 2 is higher than that at the time of induction, the subsequent prognosis is favorable.

## Supporting information

**S1 Table. Comparison of leukocyte subtype count, rate, and ratio.**
(DOCX)

**S2 Table. Comparison of biological data at week 6 between the failure and non-failure groups.**
(DOCX)

**S3 Table. Comparison of patients with clinical activity index >5 between the failure and non-failure groups.**
(DOCX)

**S4 Table. Comparison of patients treated with anti-tumor necrosis factor-α antibodies between the failure and non-failure groups.**
(DOCX)

**S1 Checklist. Human participants research checklist.**
(DOCX)

**S1 File. Review result notification.**
(PDF)

**S2 File. Review result notification (English translation).**
(PDF)

## Author Contributions

**Conceptualization:** Natsuki Ishida, Ken Sugimoto.

**Data curation:** Natsuki Ishida, Kenichi Takahashi, Yusuke Asai, Takahiro Miyazu, Satoshi Tamura, Shinya Tani, Mihoko Yamade, Yasushi Hamaya, Satoshi Osawa, Ken Sugimoto.

**Formal analysis:** Natsuki Ishida, Mihoko Yamade, Moriya Iwaizumi, Satoshi Osawa, Ken Sugimoto.

**Funding acquisition:** Natsuki Ishida, Ken Sugimoto.

**Investigation:** Natsuki Ishida, Kenichi Takahashi, Yusuke Asai, Takahiro Miyazu, Satoshi Tamura, Shinya Tani, Yasushi Hamaya, Satoshi Osawa, Ken Sugimoto.

**Methodology:** Ken Sugimoto.

**Project administration:** Natsuki Ishida, Moriya Iwaizumi, Ken Sugimoto.

**Resources:** Natsuki Ishida, Ken Sugimoto.

**Software:** Natsuki Ishida, Moriya Iwaizumi, Ken Sugimoto.

**Supervision:** Ken Sugimoto.

**Validation:** Kenichi Takahashi, Yusuke Asai, Takahiro Miyazu, Satoshi Tamura, Shinya Tani, Mihoko Yamade, Moriya Iwaizumi, Yasushi Hamaya, Satoshi Osawa, Ken Sugimoto.

**Visualization:** Ken Sugimoto.

**Writing – original draft:** Natsuki Ishida.

**Writing – review & editing:** Ken Sugimoto.

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
