## [Decision Letter · Decision Letter 0]

11 Oct 2023

PONE-D-23-29818Albumin change predicts failure in ulcerative colitis treated with adalimumabPLOS ONE

Dear Professor Sugimoto,

Thank you for submitting your manuscript to PLOS ONE. After careful consideration, we feel that it has merit but does not fully meet PLOS ONE’s publication criteria as it currently stands. Therefore, we invite you to submit a revised version of the manuscript that addresses the points raised during the review process. Your manuscript was assessed by two specialized reviewers in this field, and some major issues have been raised by both reviwers as shown below. Regarding the criticisms from both reviewers, you must fully address the issues in the revised results and discussion sections. In particular, as suggested by reviewer 1, please carefully investigate if albumin changes can predict failure in ulcerative colitis patients treated with inflixmab and golimumab as well. Without providing the additional data/explanations, this manuscript will not be considered for publication in PLOS ONE.

We look forward to receiving your revised manuscript.

Kind regards,

Emiko Mizoguchi, M.D., Ph.D.

Academic Editor

PLOS ONE

Journal Requirements:

2. No Supporting Information captions

Please include captions for your Supporting Information files at the end of your manuscript, and update any in-text citations to match accordingly. Please see our Supporting Information guidelines for more information: http://journals.plos.org/plosone/s/supporting-information.

Reviewers' comments:

Reviewer's Responses to Questions

**Comments to the Author**

1. Is the manuscript technically sound, and do the data support the conclusions?

Reviewer #1: Yes

Reviewer #2: Yes

2. Has the statistical analysis been performed appropriately and rigorously? 

Reviewer #1: Yes

Reviewer #2: Yes

3. Have the authors made all data underlying the findings in their manuscript fully available?

Reviewer #1: Yes

Reviewer #2: Yes

4. Is the manuscript presented in an intelligible fashion and written in standard English?

Reviewer #1: Yes

Reviewer #2: Yes

5. Review Comments to the Author

Reviewer #1: In this study, the authors demonstrated that serum albumin change was a prognostic factor for the therapeutic effect of ADA in UC. This is a very interesting result, but it is necessary to confirm whether similar findings can be obtained for other anti-TNF-alpha antibodies. Furthermore, it has been reported that the Neutrophil-to-Lymphocyte ratio (NLR) is associated with treatment failure of anti-TNF-alpha antibodies, which also needs to be investigated. I raise several concerns including this point as listed below.

1. Previous reports have shown that albumin changes are associated with treatment failure in UC patients using IFX and ADA. In Japan, it remains unclear whether albumin changes are associated only with ADA or with anti-TNF-alpha antibodies as a whole. Please investigate whether albumin changes can predict failure in UC patients treated with infliximab (IFX) and golimumab (GOM).

2. Factors such as Erythrocyte sedimentation rate and NLR should also be considered in ADA, IFX, and GOM.

3. Were there any differences between failures and non-failures in laboratory parameters at Week 6?

Reviewer #2: The authors aimed to investigate serum albumin (Alb) change as a prognostic factor for the therapeutic effect of ADA in UC, and demonstrated albumin change predicts failure in ulcerative colitis treated with adalimumab. The present study was well-organized and well-investigated, and will give us a new information, especially in the field of clinical medicine. To improve the quality of this paper, the authors should revise it according to the following suggestions;

1) This paper shows that the Week 2/week 0 Alb ratio is important as an outcome predictor of ADA treatment. However, the first problem is that the number of cases is small.

2) Week 2 Alb data is already data after ADA treatment, and it has already been shown that Week 2 Alb increases in the no-failure group. We believe that the Week 2/week 0 Alb ratio is not a predictive factor in a strict sense, but rather a factor for determining early treatment effects of ADA.

3) It has been reported that low serum Alb levels are factor in treatment failure in UC in general, regardless of treatment. It is predicted that this does not indicate the usefulness of the Week 2/week 0 Alb ratio only for ADA treatment. The Week 2/week 0 Alb ratio should also be indicated in patients who do not use ADA.

6. PLOS authors have the option to publish the peer review history of their article (what does this mean?). If published, this will include your full peer review and any attached files.

Reviewer #1: No

Reviewer #2: No

---

## [Author Response · Author response to Decision Letter 0]

24 Nov 2023

Emiko Mizoguchi

Academic Editor

PLOS ONE

Dear Dr. Emiko Mizoguchi:

We wish to resubmit our revised manuscript, titled “Albumin change predicts failure in ulcerative colitis treated with adalimumab.” The manuscript ID is PONE-D-23-29818.

We are grateful for the opportunity to revise our manuscript, and we would like to thank the reviewers for their helpful suggestions and comments. We have addressed the concerns of the reviewers and editor, and the relevant changes have been incorporated into the revised manuscript, which are highlighted in red font. The detailed and pointwise responses to all comments have been prepared and are given below. As advised by an academic editor, we have added an analysis of infliximab and golimumab in addition to adalimumab.

We believe that these revisions have strengthened our manuscript and hope that the revised manuscript is now suitable for publication in PLOS ONE. We have uploaded marked and unmarked copies of the manuscript, as requested.

Sincerely,

Ken Sugimoto

First Department of Medicine, Hamamatsu University School of Medicine

1-20-1 Handayama, Higashi-ku, Hamamatsu 431-3192, Japan

Tel: +81-53-435-2261

Fax: +81-53-434-9447

E-mail: sugimken@hama-med.ac.jp

 

Reviewer #1: In this study, the authors demonstrated that serum albumin change was a prognostic factor for the therapeutic effect of ADA in UC. This is a very interesting result, but it is necessary to confirm whether similar findings can be obtained for other anti-TNF-alpha antibodies. Furthermore, it has been reported that the Neutrophil-to-Lymphocyte ratio (NLR) is associated with treatment failure of anti-TNF-alpha antibodies, which also needs to be investigated. I raise several concerns including this point as listed below.

1. Previous reports have shown that albumin changes are associated with treatment failure in UC patients using IFX and ADA. In Japan, it remains unclear whether albumin changes are associated only with ADA or with anti-TNF-alpha antibodies as a whole. Please investigate whether albumin changes can predict failure in UC patients treated with infliximab (IFX) and golimumab (GOM).

Response: Thank you for your comment. As you have pointed out, past studies have examined the prediction of treatment failure of the Alb ratio of IFX and ADA. In this revision, IFX and GOM as well as the Alb ratio analysis were added. As indicated in the limitations of the original manuscript, the number of UC patients treated with ADA in this study was small. Unfortunately, the sample size was even smaller with 14 and 20 patients receiving IFX and GOM, respectively. As it was considered difficult to obtain statistically significant results for GOM and IFX separately, we added the same analysis for all anti-TNF-α antibody preparations: ADA, GOM, and IFX. The results of this analysis have been added to the newly added “Examination of the week 2/week 0 Alb ratio in patients with anti-tumor necrosis factor-α antibodies treated” section (Page 13, Lines 212-225). Of the 68 patients who received ADA, GLM, and IFX, 20 had treatment failure. Patient demographics are shown in the newly added Supplemental Table 4. The Alb ratio of the failure group was significantly lower than that of the non-failure group (P < 0.001). The cutoff value for the week 2/week 0 Alb ratio in the ROC analysis for predicting treatment failure was 0.98. Kaplan–Meier analysis was performed between the Alb ratio ≥0.98 and <0.98 groups, and the log-rank test showed a significant difference (P < 0.001).

As mentioned above, the analysis that included GLM and IFX also demonstrated the usefulness of the Alb ratio for predicting treatment failure. Furthermore, it is desirable to investigate the usefulness of the Alb ratio in the analysis of IFX and GLM, and it is hoped that more cases will be accumulated (Page 17, Line 315-319).

2. Factors such as Erythrocyte sedimentation rate and NLR should also be considered in ADA, IFX, and GOM.

Response: Thank you for your comment. We greatly appreciate the reviewers’ advice. As mentioned above, due to the small sample size, it was not possible to conduct an analysis on IFX and GOM, but we added an analysis on the leukocyte subtype. Leukocyte subtype measurements were not performed in two patients before induction, and analysis was conducted only in 32 patients (19 non-failure versus 13 failure). A significant difference test was performed on the leukocyte subtype (neutrophile, lymphocyte, and monocyte), the absolute count, and leukocyte subtype ratio (neutrophile to lymphocyte ratio [NLR], neutrophile to monocyte ratio [NMR], and lymphocyte to monocyte ratio [LMR]). There were no significant differences between the two groups in these variables. Other studies showed significant differences in NLR, but this may be due to differences in sample size and endpoints. We have presented these results in Supplemental Table 1 and also included them in the Results and Discussion section (Page 9, Line 157–159) (Page 16, Line 282–286).

3. Were there any differences between failures and non-failures in laboratory parameters at Week 6?

Response: Thank you for your comment. we compared the biological data of the failure and non-failure groups at week 6. Further exclusion of three patients with treatment failure resulted in a comparison of 10 and 21 patients with and without treatment failure respectively. There were no significant differences between the two groups in Alb, CRP, WBC, Hb, and Plt level at week 6. It is possible that cases of early treatment failure were eliminated, leaving only cases with treatment failure with relatively good data. However, the week 2/week 0 Alb ratio, which is an earlier evaluation, was more predictive of the subsequent clinical course than the week 6 data.

This result will be added to the Results section (Page 9, Lines 159-161).

Reviewer #2: The authors aimed to investigate serum albumin (Alb) change as a prognostic factor for the therapeutic effect of ADA in UC, and demonstrated albumin change predicts failure in ulcerative colitis treated with adalimumab. The present study was well-organized and well-investigated, and will give us a new information, especially in the field of clinical medicine. To improve the quality of this paper, the authors should revise it according to the following suggestions;

1) This paper shows that the Week 2/week 0 Alb ratio is important as an outcome predictor of ADA treatment. However, the first problem is that the number of cases is small.

Response: Thank you for your comment. A major problem with this study is that it had a small sample size and was barely enough to perform statistical analysis. Additionally, the fact that the albumin ratio was significantly different between the two groups despite the small sample size shows the usefulness of the albumin ratio. This comment was added in the Discussion section (Page 19, Lines 314–315).

2) Week 2 Alb data is already data after ADA treatment, and it has already been shown that Week 2 Alb increases in the no-failure group. We believe that the Week 2/week 0 Alb ratio is not a predictive factor in a strict sense, but rather a factor for determining early treatment effects of ADA.

Response: Thank you for your comment. The Alb ratio may indicate early treatment effects of ADA. However, treatment effects do not always indicate subsequent prognosis, and not all treatment effects at week 2 indicate subsequent outcomes. The results of this analysis show that the week 2/week 0 Alb ratio can be a predictive factor for treatment failure. However, the reviewer’s recognition of early treatment effects of ADA is important and has been added to the discussion section (Page 19, Lines 299–300).

3) It has been reported that low serum Alb levels are factor in treatment failure in UC in general, regardless of treatment. It is predicted that this does not indicate the usefulness of the Week 2/week 0 Alb ratio only for ADA treatment. The Week 2/week 0 Alb ratio should also be indicated in patients who do not use ADA.

Response: Thank you for your comment. The usefulness of the week 2/week 0 Alb ratio has already been demonstrated in studies on ADA and IFX. In addition to ADA, the editor and reviewer 1 also recommended analysis of IFX and GLM, but these drugs did not have a sufficient sample size to statistically analyze the data.

We will conduct another analysis if more cases are accumulated. Thus, this study focused only on ADA. Accordingly, we have added a comment to the text regarding the fact that IFX and GLM analysis could not be performed (Page 16, Lines 279–282)

---

## [Decision Letter · Decision Letter 1]

28 Nov 2023

Albumin change predicts failure in ulcerative colitis treated with adalimumab

PONE-D-23-29818R1

Dear Dr. Ken Sugimoto:

We’re pleased to inform you that your manuscript has been judged scientifically suitable for publication and will be formally accepted for publication once it meets all outstanding technical requirements.

Kind regards,

Emiko Mizoguchi, M.D., Ph.D.

Academic Editor

PLOS ONE

Additional Editor Comments (optional):

Reviewers' comments:

Reviewer's Responses to Questions

**Comments to the Author**

1. If the authors have adequately addressed your comments raised in a previous round of review and you feel that this manuscript is now acceptable for publication, you may indicate that here to bypass the “Comments to the Author” section, enter your conflict of interest statement in the “Confidential to Editor” section, and submit your "Accept" recommendation.

Reviewer #1: All comments have been addressed

Reviewer #2: All comments have been addressed

2. Is the manuscript technically sound, and do the data support the conclusions?

Reviewer #1: Yes

Reviewer #2: Yes

3. Has the statistical analysis been performed appropriately and rigorously? 

Reviewer #1: I Don't Know

Reviewer #2: Yes

4. Have the authors made all data underlying the findings in their manuscript fully available?

Reviewer #1: Yes

Reviewer #2: Yes

5. Is the manuscript presented in an intelligible fashion and written in standard English?

Reviewer #1: Yes

Reviewer #2: Yes

6. Review Comments to the Author

Reviewer #1: (No Response)

Reviewer #2: The authors responded to our suggestions and carefully revised, especially in the Discussion. We have no claim in the revised paper.

7. PLOS authors have the option to publish the peer review history of their article (what does this mean?). If published, this will include your full peer review and any attached files.

Reviewer #1: No

Reviewer #2: No

---

## [Editor Report · Acceptance letter]

20 Dec 2023

PONE-D-23-29818R1 

PLOS ONE

Dear Dr. Sugimoto, 

I'm pleased to inform you that your manuscript has been deemed suitable for publication in PLOS ONE. Congratulations! Your manuscript is now being handed over to our production team.

Kind regards, 

on behalf of

Dr. Emiko Mizoguchi 

Academic Editor

PLOS ONE